# Modulation of Reoviral Cytolysis (II): Cellular Stemness

**DOI:** 10.3390/v15071473

**Published:** 2023-06-29

**Authors:** Tarryn Bourhill, Leili Rohani, Mehul Kumar, Pinaki Bose, Derrick Rancourt, Randal N. Johnston

**Affiliations:** Department of Biochemistry and Molecular Biology, Cumming School of Medicine, University of Calgary, Calgary, AB T2N 4N1, Canada; tarryn.bourhill@ucalgary.ca (T.B.);

**Keywords:** reovirus, oncolytic virus, oncolytic viral therapy, stemness, pluripotency, cytolysis, reprogramming

## Abstract

Oncolytic viruses (OVs) are an emerging cancer therapeutic that are intended to act by selectively targeting and lysing cancerous cells and by stimulating anti-tumour immune responses, while leaving normal cells mainly unaffected. Reovirus is a well-studied OV that is undergoing advanced clinical trials and has received FDA approval in selected circumstances. However, the mechanisms governing reoviral selectivity are not well characterised despite many years of effort, including those in our accompanying paper where we characterize pathways that do not consistently modulate reoviral cytolysis. We have earlier shown that reovirus is capable of infecting and lysing both certain types of cancer cells and also cancer stem cells, and here we demonstrate its ability to also infect and kill healthy pluripotent stem cells (PSCs). This led us to hypothesize that pathways responsible for stemness may constitute a novel route for the modulation of reoviral tropism. We find that reovirus is capable of killing both murine and human embryonic and induced pluripotent stem cells. Differentiation of PSCs alters the cells’ reoviral-permissive state to a resistant one. In a breast cancer cell line that was resistant to reoviral oncolysis, induction of pluripotency programming rendered the cells permissive to cytolysis. Bioinformatic analysis indicates that expression of the Yamanaka pluripotency factors may be associated with regulating reoviral selectivity. Mechanistic insights from these studies will be useful for the advancement of reoviral oncolytic therapy.

## 1. Introduction

OVs are an exciting possible augmentation to current therapies for the treatment of cancer. This novel therapeutic consists of viruses that are either engineered or that naturally target and kill cancer cells while leaving healthy tissues mainly unharmed. The lytic replication of OVs allows for the generation and spread of progeny virus while killing the host infected cells. These viruses not only kill cancer cells directly but can also stimulate an anti-tumour immune response and thus have the potential to allow for the development of lasting immunity against particular cancers. As the therapy is targeted, it potentially allows for a greater therapeutic window and the reduction of potential side effects [1,2,3]. Reovirus is a promising OV as it is not linked to serious disease, has natural tumour tropism and is well-studied. It is undergoing phase III clinical trials and has received FDA approval for its use as an orphan drug for certain cancers [4,5,6,7,8,9,10,11,12].

The name reovirus is derived as a partial acronym for Respiratory Enteric Orphan virus [13]. The natural endemic virus preferentially infects respiratory and intestinal tissues but does not elicit severe pathology and is therefore designated an orphan virus. Reovirus is a highly stable non-enveloped virus that consists of two protein coats, an outer capsid and the inner core that surround ten dsRNA genome segments [14]. Most OVs in use have been engineered to preferentially replicate in cancers by taking advantage of the aberrant signalling that establishes the hallmarks of cancer. A fascinating property of reovirus is that it has an inherent tropism for many types of transformed cells. This unique ability to preferentially replicate in transformed cells was first demonstrated in 1977 by Hashiro et al. [15]. One year later Duncan et al. corroborated this finding by showing that reovirus would preferentially lyse transformed WI-38 lung fibroblast cells [16]. Almost 20 years later, Lee’s group demonstrated the first use of reovirus as an oncolytic agent in vivo for Ras-transformed cancer cells and showed that it could reduce tumour burden in mice [7].

Since then, reovirus has emerged as one of the most clinically advanced oncolytic therapies under development, with over thirty clinical trials (phase I-III) in numerous cancer types including colorectal, bladder, skin, head and neck, brain, non-small-cell lung cancer, ovarian, gastric, pancreatic prostate and breast [17,18]. In 2015, reovirus received orphan drug status from the FDA and European Medicine Agency (EMA) for the treatment of gastric, pancreatic and ovarian cancers [4,5,19]. Reolysin^®^ has also received fast track designation and a special protocol assessment agreement from the FDA for the treatment of metastatic breast cancer [20,21]. Even so, responses to reoviral monotherapy in advanced human cancers have in most cases been modest and short-lived, despite the promising preclinical data in animals [22], and much effort is currently focused on the testing of combination therapies. Indeed, the clinical responses to OV therapies in general have been disappointing [6,23,24], and this has led to the development of next generation OVs that are more potent, specific and safe, plus efforts to identify patients with cancers that will respond best to these therapies.

There are numerous barriers to effective clinical application of oncolytic viral therapies, some of which include inefficient delivery of viral vectors, antiviral immunity, limited vector spread within tumours and inefficient infection of tumour cells [25]. There are many strategies being employed to design the next generation of oncolytic viral vectors that overcome these barriers. The first is through arming OVs to improve viral potency. This is done by engineering viruses to express pro-drug convertases (suicide genes) or therapeutic proteins. An additional strategy for advancing oncolytic vectors is to shield them from neutralising antibodies that can inactivate viruses before they reach the tumour, through the use of chemical protective barriers, serotype switiching or the use of carrier cells. The final approach to enhancing oncolytic vectors through genetic engineering has been to improve viral specificity for tumours by the insertion of tumour or tissue specific enhancers and promoters that limit the expression of genes essential for viral replication. This approach can also be used to regulate the expression of suicide genes and therapeutic proteins [26,27,28,29].

Although attempts have been made to engineer reovirus in similar ways, it has a dsRNA segmented genome, making it challenging to engineer the virus and generate viable mutants. The first entirely plasmid based system for engineering reovirus was developed by Dermody’s group in 2007 [30]. One of the first replication competent engineered reoviruses to be produced using this system was developed by Hoeben’s group. They attached a fluorescent protein (improved Light, Oxygen or Voltage sensing domain from Arabidopsis thaliana, iLOV) into the binding region of σ1 [31]. Despite these recent advances in reoviral engineering, none of the isolated mutants has yet to progress to the clinic as an oncolytic vector. Mechanistic insights into what regulates reoviral tropism for transformed cells is critical for improving reoviral therapeutic strategies [32]. In addition, an understanding of the fundermental interactions that govern reoviral tropism will aid in the development of a panel of biomarkers that could indicate whether or not patients are likely to respond to reoviral treatment. An additional benefit of understanding reoviral selectivity would be the potential for developing rational drug combinations for cancer therapy as reovirus has so few off target effects.

However, the molecular mechanisms for reoviral selective replication in transformed cell lines are not completely understood. An early breakthrough was achieved by Lee’s group, who argued that reoviral selectivity is dependent on aberrant RAS activation and subsequent down regulation of protein kinase R (PKR) [7,12,32,33,34,35,36,37,38,39,40,41]. In healthy cells, PKR functions as part of an antiviral cellular defence system. When two of these proteins bind to dsRNA, transphosphorylation activates these kinases, which then prevent the formation of the translation initiation complexes and thereby preventing protein translation [42,43]. When the presence of dsRNA viral genomes is detected by PKR, protein synthesis is halted, which causes rapid apoptosis and viral clearance. PKR has been shown to be inactive in RAS transformed cells [36]. In cancer cells where PKR is dysfunctional and is no longer able to detect the presence of dsRNA and activate appropriate immune responses, reoviral replication is in theory uninhibited, leading to cytolysis. However, tests of this model yielded conflicting results and the exact mechanisms co-ordinating RAS activation and PKR signalling in promoting reoviral replication [35] and the general control of protein synthesis [44,45] remain unclear. Indeed, numerous groups have shown that RAS and PKR do not fully account for reovirus’ tumourigenic selectivity [46]. Contrary to the model proposed by Lee, Terasawa and colleagues were able to show that some cancers with low RAS activity are unexpectedly susceptible to reoviral oncolysis, while others with high RAS activation were resistant [46] (see also our accompanying paper). Knockdown of RAS did not prevent reoviral replication in colorectal cancer cell lines [47]. Smakman et al. subsequently showed that deletion of KRAS through homologous recombination did not prevent reoviral replication [48]. In accordance with these findings, work by van Holt et al. showed that cell lines or tissue fragments taken from colorectal tumours were non-permissive to reovirus regardless of KRAS activation [49]. Similarly, in work focusing on PKR activation Zhang et al. demonstrated that siRNA knockdown of PKR expression did not increase reoviral lysis in tumour cells [50]. Work presented by Twigger et al. also showed that inhibiting PKR activity in resistant cell lines did not cause an increase in reoviral response [51]. Schiff’s group presented data entirely contrary to Lee’s work showing that the presence of either RNAse L or PKR modulated reoviral replication [52]. This team subsequently verified their findings by showing that reovirus replication was more efficient in PKR knockout cells [53]. The debate surrounding the RAS/PKR pathway clearly indicates viral tropism is a complex process involving numerous integrated signalling networks.

The interactions between host and viral signaling that determine successful reoviral oncolysis are likely to be complex and varied. Efficient receptor targeting and binding, viral replication, type of virally induced programmed cell death (autophagy, apoptosis, necrosis, pyroptosis) and the susceptibility of a cancer to different cell death programs, as well as host cell immune responses, all influence the efficiency of oncolysis [23,32,54,55,56]. As our understanding of what governs reoviral selectivity in transformed cells is incomplete, there remain unique opportunities to uncover these mechanisms and apply this knowledge to enhance reoviral oncolytic efficiency.

Remarkably, reovirus not only has anti-tumour activity but its selective cytolysis extends to cancer stem cells (CSCs) [57,58]. This is intriguing as both bulk tumour cells and CSCs exist in a de-differentiated state relative to their healthy counterparts. What is also striking about reoviral infection is that is capable of lysing healthy, non-cancerous stem cells. An earlier observation from our lab demonstrated that murine embryonic stem cells are sensitive to reovirus both in vitro and in vivo. We found that wild type reovirus could reduce in SCID mice the size of teratomas that were derived from non-cancerous murine embryonic stem cells [59]. Reovirus can therefore selectively lyse bulk cancer cells, cancer stem cells and healthy murine embryonic stem cells, all of which have stem-like characteristics. In addition, healthy differentiated tissues are mostly refractory to infection. These observations led us to speculate that pathways responsible for stemness may govern some aspects of reoviral selectivity. To test this hypothesis we began by characterizing the capabilities of reovirus to lyse non-cancerous stem cells. We then investigated whether inducing pluripotency and differentiation could alter the cellular responses to reoviral infection. Finally, we performed bioinformatic analysis to understand the potential role of Yamanaka stem cell factors in regulating reoviral selectivity.

An appreciation for the signalling pathways responsible for governing reoviral tropism will be a foundation for improving the efficacy of this oncolytic agent. Findings from this study could help establish which cancer backgrounds are amenable to cytolysis, information that will be critical for developing effective biomarkers to screen patients. Additionally, knowledge of reoviral selectivity can aid in the development of synergistic therapeutic combinations enhancing potency. Mechanistic insights on the signalling pathways regulating viral tropism will enable the development of next generation reoviral oncolytic therapies.

## 2. Materials and Methods

### 2.1. Cancer Cell Lines and Culture Conditions

Cancer cells were grown in high glucose (25 mM) Dulbecco’s Modified Eagle Medium (DMEM) supplemented with 10% (*v*/*v*) foetal bovine serum (FBS) (Thermo Fisher Scientific, Waltham, MA, USA) and 100 units/mL penicillin and 100 mg/mL streptomycin (Thermo Fisher Scientific). Cell lines cultured in this medium formulation included: L929 (murine fibroblasts, American Type Culture Collection, ATCC, Manassas, VA, USA), Hela (human cervical adenocarcinoma, ATCC), ZR-30–75 (human mammary ductal carcinoma, ATCC), MDA-MB-231 (human mammary adenocarcinoma, ATCC), MDA-MB-468 (human mammary adenocarcinoma, ATCC), MCF7 (human mammary adenocarcinoma, ATCC) and EMT6 (murine mammary carcinoma, ATCC). All cell lines used gave broadly similar responses, but only experiments performed with cells in triplicate or greater are presented in the Results. The N_2_O_2_ (mammary tumour cells from MMTV-Neu mice) cell line was generated by DR [60]. Cells were grown in monolayers and incubated in a humidified atmosphere of 5% CO2 with 95% humidity at 37 °C. When cultures reached 70–80% confluency they were passaged using Trypsin-ethylenediaminetetraacetic acid (EDTA), (Thermo Fisher Scientific; 0.25% Trypsin, 0.913 mM EDTA). Trypsinized cells were incubated with fresh medium and counted using Trypan blue (Thermo Fisher Scientific) and a haemocytometer. Cells were then placed in fresh medium and seeded onto appropriate culture dishes. Stocks of these cell lines were maintained in liquid nitrogen for long term storage in a 90% FBS 10% Dimethyl sulfoxide (DMSO, Sigma, St. Louis, MO, USA) solution.

Induced pluripotent cancer stem cells (14ex-11, Yulia Shulga, Rancourt lab) were grown in high glucose (25 mM) DMEM medium supplemented with 15% (*v*/*v*) embryonic stem cell FBS (Thermo Fisher Scientific), 1 mM sodium pyruvate (Thermo Fisher Scientific), 0.1 mM non-essential amino acids (Thermo Fisher Scientific), 0.1 mM 2-mercaptoethanol (Thermo Fisher Scientific) and 10 ng/mL leukaemia inhibitory factor (LIF, Sigma). These cells were grown on a feeder cell layer of inactivated murine embryonic fibroblasts

Induced pluripotent cancer stem cells were generated by reprogramming N_2_O_2_ tumour cell lines with PiggyBac transposon vectors that expressed cDNA for OCT4, c-MYC, KLF4 and SOX2 [61]. Reprogramming was confirmed through qRT-PCR and immunofluorescence imaging for stem cell markers. Pluripotency of this reprogrammed cell line was additionally confirmed via in vivo teratoma assay.

### 2.2. Stem Cell Culture Conditions

#### 2.2.1. Murine Cell Lines

Murine embryonic (D3) and induced pluripotent iPS-3, stem cells were cultured in high glucose (25 mM)) DMEM that was supplemented with; 15% (*v*/*v*) embryonic stem cell FBS, 1 mM sodium pyruvate, 0.1 mM non-essential amino acid, 0.1 mM 2-mercaptoethanol and 10 ng/mL LIF. Both cell lines were provided by the Rancourt lab [62]. Stem cells were co-cultured on inactivated murine embryonic fibroblast cells, as described next.

Murine embryonic fibroblast (MEF) cells were cultured on 0.1% gelatin (Sigma) coated tissue culture plates. These cells were grown in high glucose (25 mM) DMEM medium with 10% (*v*/*v*) FBS, 0.1 mM non-essential amino acids, 50 units/mL penicillin and 50 mg/mL streptomycin. MEFs were used for experimental purposes and as a feeder layer for stem-like cells. MEFs were prepared as a feeder layer and when they reached 100% confluency the cells were treated with 10 µg/mL mitomycin C (Sigma) for 2 h at 37 C. The cells were then washed with PBS and incubated in fresh medium overnight. Inactivated MEFs were used as feeder cells the following day [63]. When stem cells were required for experiments they were removed from MEFs. Once the stem cells reach 70–80% confluency on the gelatin plates the cells were dissociated and counted using a haemocytometer and Trypan blue. The cells were then aliquoted in appropriate amounts for subsequent experiments. Stocks of these cell lines were maintained in liquid nitrogen for long term storage in a 90% FBS 10% DMSO solution.

#### 2.2.2. Human Stem Cells

Human embryonic stem cell lines H9 (Wicell, Madison, WI, USA), H1 (Wicell, WA01) and induced pluripotent human stem cells BJ-EOS clone 4YA (provided by the Ellis lab) [64] were cultured in mTeSR complete medium (Stem Cell Technologies, Vancouver, Canada) supplemented with 50 units/mL penicillin and 50 mg/mL streptomycin. These cells were cultured on Matrigel (Corning, New York, NY, USA) treated 60 mm plates as per the manufacturers specifications [64].

Cells were passaged by washing the cultures with Dulbecco’s phosphate-buffered saline without calcium and magnesium (DPBS, Stem Cell Technologies) twice followed by accutase (Stem Cell Technologies) treatment. Once the cells were in a single cell suspension fresh medium was used to inhibit accutase activity. Cells were then counted and aliquoted for subsequent experiments. Cell were maintained in medium containing 10 μM Rho-associated kinase (ROCK) inhibitor (Y-27632; Stem Cell Technologies) for 24 h after passaging and then returned to their regular medium.

Human foreskin fibroblast cells (HFF) [64] were cultured on gelatin coated plates in high glucose (25 mM) DMEM medium supplemented with; 15% (*v*/*v*) embryonic stem cell FBS, 1.0 mM sodium pyruvate, 0.1 mM non-essential amino acid and 0.1 mM 2-mercaptoethanol.

### 2.3. Differentiation Protocols

#### 2.3.1. Murine Cell Lines

Murine embryonic and induced stem cells were differentiated through the hanging drop technique [63,65,66]. Trypsinized stem cells were resuspended in medium that did not contain LIF [high glucose (25 mM) DMEM that was supplemented with 15% (*v*/*v*) embryonic stem cell FBS, 1.0 mM sodium pyruvate, 0.1 mM non-essential amino acids and 0.1 mM 2-mercaptoethanol]. Cells were counted and diluted to a concentration of 25,000 cells/mL (or 500 cells per 20 μL drop). Drops of 20 μL were plated onto the lid of a 90-mm bacterial culture plate. The lid was then placed on top of a plate that contained 10 mL of PBS. Hanging drops were incubated at 37 C in 5% CO_2_, with 95% humidity for a period of 3–4 days [63]. Embryoid bodies were then transferred into 24, 48 or 96 well cell culture plates (Greiner Bio-One, #662160, Kremsmünster, Austria) that were treated with 0.1% gelatin. The embryoid bodies were either infected with reovirus directly as the cells were being transferred into the multiwell plates or after 1 or 10 days of differentiation. Spontaneous differentiation was confirmed by the presence of beating cardiomyocytes in cultures. Beating cells were identified through the use of an Olympus CKX41 inverted phase contrast microscope.

#### 2.3.2. Human Cell Lines

Human embryonic stem cell (hESC) line H1 (WiCell) and human induced pluripotent stem cell (hiPSC) line 4YA (generated from BJ cell lines in Dr. James Ellis’ laboratory at the University of Toronto) were differentiated into liver organoids using a modified previously published protocol [67]. The differentiation process was performed through three stepwise stages: definitive endoderm (DE) formation, hepatoblast, and mature hepatocytes.

Briefly, human pluripotent stem cells (hPSCs) were seeded on matrigel-coated 60-mm plates at a density of 4 × 10^5^ cells per plate, and when the cells reached to 90% confluency the directed differentiation toward definitive endoderm was started for three days. The first step of differentiation (DE) was done on 2D culture plates. The cells were treated with RPMI 1640 medium (Thermo Fisher scientific) containing 1% BSA (Sigma-Aldrich, St. Louis, MO, USA), 1% B27 (Thermo Fisher Scientific). On day 1 this medium was supplemented with 6 mM CHIR99021 (Stem Cell Technologies), and following this the medium was supplemented with 10 ng/mL activin A (R&D Systems, Minneapolis, MN, USA) on days 2 and 3 of the differentiation protocol.

Stage two of differentiation (hepatoblast) began with aggregating putative endoderm cells into aggregates using AggreWell 400 24-well plates. Aggregates containing 100 cells were used for spheroid formation. To calculate the number of cells needed in each well of AggreWell 400 24-well plate, the following formula was used:1200 microwells per well (24 wp) = 1200 aggregate × # cells/aggregate

The endodermal cells were dissociated using Accutase and collected into fresh stage two medium containing DMEM-F12 medium (Thermo Fisher Scientific) supplemented with 2% knockout serum replacement (KOSR, Thermo Fisher Scientific), 10 ng/mL hepatocyte growth factor, (HGF, R&D Systems), and 10 ng/mL fibroblast growth factor-4 (FGF4, R&D Systems). Cells were aggregated by adding 600 µL cell suspension to each well of a 24-well plate containing 10 µM ROCK inhibitor (Y27632, Stem Cell Technologies) and placed in the incubator for 24 h. The next day, aggregates were placed into fresh stage two medium in 6-well plates (non-TC treated) on a shaker (f < 60 rpm). Aggregates were then treated with stage two differentiation medium for a period of 7 days on the shaker.

Formed organoids were then subjected to the third and final stage of differentiation (mature hepatocytes) where they were treated for a period of 10 days using HCMTM Hepatocyte Culture Medium BulletKitTM (Lonza, Basel, Switzerland) supplemented with 0.1 mM DMSO, 1 mM non-essential amino acids, 1 mM GlutaMAX and 10 ng/mL oncostatin M (R&D Systems) [67]. Stage 3 hepatocyte-like differentiation liver organoids were then subject to reoviral infection. Differentiation of the liver organoids was confirmed through immunofluorescence imaging of liver specific markers KRT7, CYP3A4, HNF4-alpha and Sox9.

#### 2.3.3. Reoviral Infection

Reovirus type 3 Dearing strain was used for all infections and was obtained from Dr. Patrick Lee (Dalhousie University, Canada). To determine concentration and stability of reovirus, all batches were extensively titred by plaque assay in L929 cells [33].

### 2.4. Viability Assays

To determine cell survival after reoviral infection, crystal violet staining was used to quantify viable cells that adhered to the multi-well plate surface [68,69]. For viability assays reoviral infection took place one day after adherent cells were seeded into 24 well plates. Reoviral infection of murine embryoid bodies took place immediately as embryoid bodies were seeded into plates, or 10 days after seeding. In the case of human liver organoids, these were infected immediately or one day after seeding. A standard range of MOIs (multiplicity of infection or the ratio of plaque forming units, viral particles, per cell) from 1–100 was used. Reovirus was added to wells in a desired final volume and cells were left on a rocking platform at room temperature for 30 min. Infected cells were then placed in an incubator (at 37 °C with 5% CO_2_ with 95% humidity) and left until the experimental endpoint was reached. Viability assays were conducted for a period of three days.

After reoviral infection medium was removed from the cells, and they were then fixed to the plate through the addition of 4% paraformaldehyde (PFA, Sigma) solution. The fixative solution was removed and plates allowed to air dry. Subsequently a 1% crystal violet (Sigma) solution was added to each well and plates incubated at room temperature for 10 min. The plates were then washed with water and allowed to air dry. The crystal violet stains were dissolved in a 1% sodium dodecyl sulfate (SDS, Sigma) solution. The optical density of each well was measured at 570 nm using a microplate reader. Experiments were conducted in replicates of three, and data were normalised to an untreated control that had 100% survival. This method resulted in low levels of residual staining of cellular debris even when all cells were killed, as verified by microscopy.

### 2.5. Reovirus Protein Synthesis

#### Immunostaining and Widefield Microscopy

Reoviral protein abundance and distribution were monitored through the use of a polyclonal rabbit anti-reovirus (serotype 3 Dearing) antibody serum [70]. This primary antibody was used in both Western Blot analysis and immunostaining. Adherent cell lines (cancer and stem cell) were cultured on chambered coverslips (Ibidi µ-Slide 8 Well). Cells were seeded at 100,000 cells per chamber and subsequently infected (MOI 1-100) one day later. Cells were usually fixed one day post-infection. In the case of embryoid bodies and liver organoids, cells were fixed two and three days post infection, respectively. All cells were washed with DPBS and then fixed by adding 200 μL of 4% PFA to each well. The plates were incubated at room temperature for 15 min. The cells were then washed with DPBS and permeabilised with 100% methanol and left to incubate for 8 min at −20 C. All slides were rinsed with DPBS twice and then incubated in a 5% bovine serum albumin (BSA, Sigma) solution in DPBS for 45 min at room temperature on a rocking platform. Primary antibody (polyclonal rabbit anti-reovirus, 1 in 5000 dilution) was added to the cells for a period of 2 h. Cells were then rinsed three times in DPBS. Following this a secondary antibody (donkey anti-rabbit-Alex647, Thermo Fisher Scientific 1:5000) solution was added to the cells for one hour at room temperature. Plates were covered in foil from this point onwards. Plates were then washed with DPBS and stained with 4′,6-diamidino-2-phenylindole (DAPI; Thermo Fisher Scientific) solution. Plates were incubated for 5 min on a rocking platform and rinsed with DPBS twice. Cells were stored in the dark at 4 °C until imaged on an Olympus IX71 Evolve wide-field microscope [71]. Image capture and analysis was performed using Volocity version 6.5.1.

### 2.6. Western Blots

After cells had been infected (MOI 1-10) they were lysed and proteins collected for Western blot analysis. Cells were lysed using 120 μL lysis buffer per well (12-well plate, 4 × 10^5^ cells). Lysis buffer consisted of 1 M Tris-base (VWR), 8.75% glycerol, 10% SDS, 150 mM sodium chloride (Sigma) and 1% TritonX100 (Sigma). Protease inhibitor complex (Sigma) and 2 mM phenylmethylsulfonyl fluoride (PMSF, Sigma) was added fresh for every lysis reaction. Lysis buffer was added to the adherent cells and plates were placed on ice for 5 min while spent medium were centrifuged at 1000× *g* for 5 min. Lysates from the plates were collected and placed on pellets from the centrifuged medium. The combined cell lysates were then heated at 90 °C for 10 min.

Protein samples were then quantified using Detergent Compatible (DC) Protein Assay Kit (Bio-Rad). BSA was used for the standard curve and reactions conducted in 96-well format. A microplate reader was used to quantify the optical density of each sample at 750 nm.

Protein samples were then separated using sodium dodecyl sulphate-poly acrylamide gel electrophoresis (SDS-PAGE) with stacking gels at 5% while resolving gels were 12%. Proteins were transferred to nitrocellulose membranes and were incubated with primary antibodies (Appendix A) overnight on a rocker at 4 C. Membranes were rinsed with TBST three times and incubated with appropriate (species specific) secondary antibodies that were conjugated to horseradish peroxidase (HRP). Membranes were washed four times in TBST and Amersham ECL Prime Western Blotting Detection Reagent (GE Healthcare, Chicago, IL, USA) was added to each membrane. Proteins were detected using the Bio-Rad ChemiDoc Touch Imaging system. Images of the membranes were captured and analysed using Image Lab version 5.2.

### 2.7. Bioinformatic Analysis

#### 2.7.1. Analysis of Reoviral Sensitive vs. Resistant Cell Lines

To determine the extent of Yamanaka factor expression within reoviral resistant vs. sensitive cancer cell lines, a literature search was conducted in PubMed to compile a dataset of relevant cell lines. Cell lines were designated sensitive if cytopathic effects were noted after infection with an MOI 20 or lower. Robust multichip average (RMA) normalised expression data for 27 sensitive and 12 resistant cell lines from the Genomics of Drug Sensitivity in Cancer database were available for analysis [72]. These data were then collated and analysed using the R programming language.

A linear model for microarray data analysis (Limma) was performed to compare resistant and sensitive cell lines to determine differential gene expression profiles [73]. Four genes (SNAP25, PON2, PRCD, C9orf16) were determined to be differentially expressed (false discovery rate 0.1). Literature review revealed PON2 was implicated as a having a role in regulating stemness and as a result genes involved in regulating PON2 expression were investigated. Spearman correlation analysis was used to investigate the relationships between the continuous gene expression of PON2 and STAT5/KLF4. Spearman correlations were visualised using scatter plots.

#### 2.7.2. Analysis of Gene Expression in Tumours

To further our analysis of the Yamanaka factors possibly involved in tumours, expression datasets from nearly 10,000 samples from 30 different cancers from The Cancer Genome Atlas (TCGA) were analysed [74]. Expression of the Yamanaka factors within the breast cancer subtype (BRCA) in the TCGA were log 2 transformed and Z-scores normalised to plot heat maps of expression. Hierarchical clustering was performed using Euclidean distancing. Spearman correlation coefficients were calculated for expression of OCT4, KLF4, SOX2, and MYC. A similar analysis was conducted in only breast cancer tumours with 1092 samples from the TCGA.

### 2.8. Statistical Analysis

Experimental raw data were normalised and maintained in Microsoft Excel. For viability assays infected cells (at various MOIs) were only compared to the untreated controls to determine whether a significant difference in the means was present. Two-tailed unpaired student’s *t*-tests were used to compare treated groups to untreated controls, with *p* values of significance set at 0.05. Data presented show standard error of the mean. Inkscape version 0.92 was used to produce graphs and visualisations of the data.

## 3. Results

In our earlier studies of an attenuated reoviral strain, we noted that wild type reovirus (Dearing strain) was surprisingly capable of infecting and killing healthy (non-cancerous) murine embryonic stem cells (mESCs), despite previous assumptions that reovirus causes little or no harm to non-cancerous cells [59]. We then showed that wild type reovirus was capable of inhibiting the growth in SCID mice of teratomas that normally arise after sub-cutaneous mESC transplantation [59]. The mechanism for why reovirus can infect some cancers and not others remains undefined and we speculated that an investigation into the similarities between stem cells and cancer cells provides a new avenue of investigation into what regulates this phenomenon. Although sensitivity of adult stem cells to reovirus has been investigated previously, particularly as groups are interested in using mesenchymal stem cells as carrier cells to deliver virus [75,76], no studies have investigated the sensitivity of pluripotent stem cells to reovirus. The first objective of this work was therefore to investigate both murine and human stem cell responses to reoviral infection.

### 3.1. Characterisation of Reoviral Cytolysis in Embryonic Stem Cells

To determine the extent to which murine ESCs were sensitive to reoviral infection, D3 embryonic stem cells were infected with reovirus with various multiplicities of infection (MOIs). L929 cells were used as a positive control in these experiments as these cells are known to be susceptible to reoviral infection and lysis (Figure 1A). It is clear that mESCs are exquisitely sensitive to reoviral infection and cytolysis when infected with MOIs of one (Figure 1A). To investigate whether these observations were restricted to murine cell types or if they were applicable to the human cells, similar experiments were performed in human embryonic stem cells lines H9 (hESCs). HeLa cells were used as a positive control. It was clear that human embryonic stem cells are also highly sensitive to reoviral infection (Figure 1B and Appendix A). To establish if viral replication was occurring within mESCs and hESCs, Western blot analysis and immunofluorescence imaging were performed. Antibodies directed towards the capsid proteins of the virus were utilized to determine whether viral proteins were being produced, which is indicative of viral replication [70]. Western blots demonstrate that reovirus can effectively replicate in mESCs and hESCs (Figure 1C,D). These results were in accordance with the immunofluorescence imaging which showed that mESCs and hESCs support active production of reoviral particles (Figure 1E,F).

It is clear from these results that healthy, non-cancerous embryonic stem cells are highly susceptible to reoviral infection and cytolysis. Both mouse and human cell lines appear to be highly sensitive to infection with low concentrations of virus producing massive cell death within the population. Likewise, both murine and human embryonic stem cells are capable of supporting reoviral replication. Cellular machinery is recruited to produce viral capsid proteins, which is indicative of active viral replication. The observation that reovirus can infect and kill non-cancerous stem cells is fascinating as it has implications for its mechanism of selectivity. Cancerous cells are thought to either arise from stem cells that have undergone oncogenic mutation and transformation or from differentiated cells that have undergone a process of reversion to a more stem-like state. We infer these responses reflect similarities between cancer cells and embryonic cells, especially in responses to cytotoxic agents such as viruses. The differentiation status of the cell may influence embryonic and cancer cells’ permissiveness to infection.

### 3.2. Characterisation of Reoviral Cytolysis in Induced Pluripotent Stem Cells

iPSCs offer a unique model in which to investigate the role that pluripotency plays in regulating cellular susceptibility to reoviral infection. These cells begin in a differentiated state and progress towards pluripotency upon introduction of the Yamanaka factors. This provides the opportunity to investigate whether a cell that is initially resistant to reovirus can be made susceptible by inducing a pluripotent state within the cell.

Murine embryonic fibroblast (MEF) cells are a mature cell type that is derived from connective tissue of the skeletal muscle of new-born mice and are frequently used to generate murine induced pluripotent stem cells (miPSC). In our lab, MEFs were transduced with lentiviral vectors carrying a payload of cDNA that encoded the four Yamanaka factors to produce a stable iPSC cell line [62]. Similarly, human induced pluripotent stem cells (hiPSC, 4YA cell line) were derived from human foreskin fibroblast (HFF) cells and were transduced with a lentiviral vector that stably integrates into genomic DNA and overexpresses the Yamanaka factors [64,77]. To determine whether the differentiation status influences reoviral selectivity, the differentiated cells (MEF and HFF) as well as the established induced pluripotent stem cells (miPSC and 4YA) were infected with reovirus. Sensitivity to the virus was subsequently determined through cell viability assays, Western blots and fluorescent imaging (Figure 2). Both murine and human induced pluripotent cells in both murine and human models are sensitive to infection and subsequent cell death (Figure 2A,B). The sensitivity was similar to that seen in embryonic stem cells. The more mature ancestral cell types however were resistant to cytolytic effects from the virus. Both Western blot analysis and immunofluorescence imaging confirmed the production of new viral capsid proteins one day after infection (MOI 10) in the pluripotent cells while differentiated cell types did not produce viral capsids proteins, indicating a lack of viral replication (Figure 2C–F).

What is particularly interesting about the results presented here is that upon induction of pluripotency programs there is a shift in reoviral sensitivity from a resistant phenotype to that of a sensitive one. The sensitivity of the induced pluripotent cells is similar to that of their embryonic counter parts with murine cells being the most sensitive while human cells are slightly less so. The fact that cells become susceptible to infection upon reprogramming is a novel finding with intriguing implications, not only for reoviral biology but also for oncogenesis. Reprogramming appears to promote reoviral susceptibility, implicating the Yamanaka factors or their downstream targets in determining cellular permissiveness to reoviral replication. We may infer that pluripotency induction mimics certain aspects of oncogenic progression. The Yamanaka factors that are required for reprogramming are often upregulated or are overexpressed in a number of cancers [78]. A good example of this is the Yamanaka factor MYC, which is a well-known oncogene and accounts for a great deal of transcriptional similarity between iPSCs and cancer cells. It is intriguing then to speculate on what role, if any, the Yamanaka factors may have on reoviral replication.

### 3.3. Reoviral Cytolysis in Differentiated Cells

It is evident that induction of pluripotency can promote characteristics of a cell that make it susceptible to reovirus. It would therefore be important to know if stem cells (induced or embryonic) could be made refractory to reoviral infection through the loss of pluripotency by promoting differentiation.

In the murine PSC context, spontaneous differentiation can be induced through the removal of leukaemia inhibitory factor (LIF) from culture media. To promote differentiation, a hanging drop technique was used to form 3D spheres known as embryoid bodies, which are reflective of spontaneous cell differentiation [65,66]. Murine pluripotent stem cells (induced and embryonic) were induced to differentiate in hanging drops for a period of three days. Following this embryoid bodies were transferred to gelatin coated plates and allowed to differentiate. Time course analysis was performed and differentiated cells were infected immediately after their hanging drop treatment as they were transferred to gelatin plates (Figure 3A), or they were allowed to differentiate for a period of five or ten days following seeding. Beating cardiomyocytes after five days indicated that differentiation had occurred in some of the cells. Cells were infected with reovirus and crystal violet viability assays performed. Cells that had undergone differentiation were refractory to reovirus as there was very little change in their cell viability particularly when compared to their stem cell counter parts (Figure 1A and Figure 2A). Surprisingly, this switch from susceptible cell type to refractory occurred relatively soon in the differentiation process where refractory cells were noted only three days after removal of LIF and suspension in hanging drops. Differentiated murine cells were then subject to immunofluorescence imaging and Western blot to determine if reoviral replication occurred within these cell types. Reovirus cann produce capsid protein as evidenced through Western blot analysis (Figure 3C). The amount of protein produced however is quite low relative to the susceptible L929 control cell lines. This trend was mirrored in the immunofluorescence images: some cells within the population appear to support reoviral replication, while the large majority of cells appear to be resistant to infection and concomitant production of capsid proteins (Figure 3G). The amount of infection seen in these cells is lower than in their pluripotent counterparts, particularly as the immunofluorescence in the differentiated cells was measured two days post infection. It appears that cells along the periphery of the embryoid bodies are infected and produce actively replicating virus. In general however, the differentiated cell populations produced much lower levels of reoviral protein when compared to susceptible controls and their stem cell counterparts.

In the case of human stem cells a method for directed differentiation was used. Cells were again encouraged to form 3D spheres in the Aggrewell system. These aggregates were then exposed to a variety of growth factors such as Activin A, Hepactocyte Growth Factor and Fibroblast Growth Factor 4 to promote endodermal differentiation [67]. Differentiation in human cells was allowed to proceed to stage three of the differentiation protocol where liver organoids are produced. Organoids were characterised using immunofluorescence imaging and the expression of liver-specific markers such as KRT7, CYP3A4, HNF4-α and Sox9 was confirmed (Appendix A). hESCs (H1) cells and hiPSCs (4YA) organoids were infected with reovirus and three days post-infection cell viability was determined. The cell viability was not reduced upon infection (Figure 3B) and was much higher when compared to stem cell populations (Figure 1B, Figure 2B and Appendix A). As organoids can vary in size and shape this creates variability in the data. To reconcile this difficulty additional replicates (H1 n = 6 and 4YA n = 9) were included in these experiments. To determine if reovirus could replicate within the organoids Western blotting and immunofluorescence imaging were employed. Western blots of the hESC organoids showed a relatively high production of reoviral proteins, especially when compared to embryonic stem cells (Figure 3E). This was unexpected and the result was not echoed in the immunofluorescence images captured from organoids where very little active reoviral replication was seen (Figure 3H). In differentiated hiPSCs organoids some reoviral protein was detectable on the Western blot, although it was much less than the protein production seen in hiPSCs (Figure 3F). Fluorescent imaging showed a similar situation where some of the peripheral cells support reoviral replication but the vast majority appeared refractory (Figure 3H).

In these experiments, murine and human stem cells become refractory to reoviral infection after differentiation has taken place. In the case of murine cells this switch from susceptible to refractory occurs relatively soon in the differentiation process, occurring three days after LIF removal. These differentiated cells appear to be resistant to reoviral lysis and there is a reduction in the amount of reoviral replication that is supported by the cells. Reoviral capsid proteins are still observed in cells after differentiation occurs. The production of reoviral proteins in some cells within differentiated cell populations is indicative of replication. This is perhaps not surprising given that differentiation is not a synchronous process and produces a population of heterogeneous cell types. The cells that sustain reoviral replication may be slower to differentiate and more stem-like in character. This may explain why only a few cells within the population support replication. This trend is similar in the differentiated hiPSCs (4YA) where cytolysis and reoviral replication are not supported after differentiation. This is in contrast to the differentiated hESCs (H1) where Western blots indicate that reoviral replication occurs after differentiation has occurred but the immunofluorescence images indicate that virus replication within this cell population is limited. From these results it appears that pluripotency can modulate reoviral sensitivity within healthy cells. This leads to a bigger question of whether cellular stemness varies within cancers such that the de-differentiated state within some cancers and not others contributes to reoviral selectivity.

### 3.4. Induced Pluripotent Cancer Stem Cell Sensitivity to Reovirus

To investigate whether pluripotency programming in cancer cells could modulate reoviral susceptibility, resistant mammary tumour cells were induced to become pluripotent through the use of PiggyBac transposon vectors carrying the Yamanaka factors. Once these induced pluripotent cancer stem cells were fully characterised, they were subjected to reoviral infection and their sensitivity was investigated.

To investigate the role of pluripotency in regulating reoviral selectivity in the context of a cancer, induced pluripotent cancer stem cells (ipCSC) were generated. A murine cell line (N_2_O_2_) derived from mammary tumours in transgenic female mice of the MMTV-Neu strain was tested for its sensitivity to reoviral cytolysis [60,79]. It was found that these cancerous cells were resistant to reoviral lysis (Figure 4A). The N_2_O_2_ cells were reprogrammed using a PiggyBac vector that expressed all four Yamanaka factors. Resulting ipCSCs (cell line 14ex-11) were characterised using immunofluorescence staining to detect OCT4, Nanog and SSEA-1 expression (Appendix A). Teratoma assays were used to determine the differentiation potential of these cells and histochemical analysis revealed the formation of all three germ layers indicating pluripotent potential in these cells (Appendix A). IpCSCs were subjected to reoviral infection and subsequent cell viability was determined (Figure 4A). The induction of pluripotency within cancer cells enhanced reoviral susceptibility and transitioned the cells from a resistant phenotype into a sensitive one. Western blot and immunofluorescence imaging assays were conducted to determine the extent to which reoviral replication occurs within parental cancer cells (N_2_O_2_) and ipCSCs. The ipCSCs and sensitive control L929 both produce reoviral capsid after initial infection while the N_2_O_2_ tumour cells were unable to support active replication (Figure 4B). Fluorescent imaging confirmed this finding showing that ipCSCs produced large amounts of reoviral protein while N_2_O_2_ cells did not (Figure 4C).

These results show that reoviral sensitivity can be introduced into an initially resistant breast cancer cell line through the induction of pluripotency programming. This is a compelling finding as reoviral sensitivity has only been induced in cancers previously through the activation of the RAS pathway. It is evident that pluripotency programming creates a cellular environment that is conducive to reoviral infection, replication and lysis. This strongly implicates reprogramming and pluripotency in the regulation of reoviral tropism and particularly, it points to the involvement of the Yamanaka factors in the mechanisms for regulating reoviral selectivity.

### 3.5. Bioinformatic Analysis of the Yamanaka Factors and Their Potential Role in Regulating Reoviral Sensitivity

To investigate the pathways responsible for cellular susceptibility to reovirus, analysis of the Yamanaka factors and the pathways they regulate was conducted, as one or a combination of these factors may influence cellular responsiveness to virus.

To begin the bioinformatic investigation into the possible pathways involved in regulating reoviral selection, a dataset with information regarding the sensitivity of cancer cell lines to reovirus was created. Cell lines were identified by review of published analyses of reoviral responses and deemed sensitive to reovirus if cytopathic effects were observed after infection at MOI 20 or lower. A data set of 106 cell lines confirmed as sensitive or resistant to virus was compiled, and among these normalised expression (RMA) data were found for 27 sensitive and 10 resistant cell lines in the Genomics of Drug Sensitivity in Cancer database.

The analysis began by assessing differentially expressed genes in sensitive and resistant cell types. Differential expression was analysed using linear models for microarray data (limma) analysis to compare the resistant and sensitive cell lines. Four candidate genes were identified, namely *SNAP25, PON2, PRCD* and *C9orf16*. *PON2* showed promise as a candidate for further investigation, as lower expression levels of this gene are associated with differentiation. In *PON2* knockouts, mice have an increase in the fraction of differentiated stem cells [80]. *PON2* gene expression was significantly upregulated in reoviral sensitive cells (Figure 5A) which may be indicative of the stem-like nature of these cell types. STAT5B directly modulates PON2 activity and there is a significant negative association between PON2 and STAT5B expression in our cell lines of interest (Figure 5B) [81]. Interestingly, there was a trend toward down regulated STAT5B expression within reoviral sensitive cells lines (Figure 5C). This led to an investigation into the relationship between STAT5B, PON2 and the Yamanaka factors within the cell lines of interest. Of the four factors investigated KLF4 showed a significant negative correlation with STAT5B expression and a positive one with PON2 expression (Figure 5D,E). These associations provide intriguing new leads for further research and require further experimental validation.

To further the analysis of the potential involvement of Yamanaka factors in reoviral selectivity, expression data from TCGA were analysed to determine patterns of expression in cancers derived from patient samples. A heat map generated using data from TCGA (BRCA- breast cancers) showed that OCT4 and MYC were overexpressed in basal-like breast cancers while SOX2 and KLF4 were down regulated in this subtype (Figure 5F). A multivariate linear analysis was performed and indicated that OCT4, MYC and KLF4 were independently and significantly associated with the basal subtype. While OCT4 (t = 27.175; *p* < 2 × 10^−16^) and MYC (t = 7.807; *p* = 1.38 × 10^−14^) had a positive association, KLF4 (t = −6.698; *p* = 3.39 × 10^−11^) had a negative association.

The four Yamanaka factors do not function in isolation and they have specific patterns of expression that regulate reprogramming and induction of pluripotency. To determine if there was a correlation among the expression levels of the four Yamanaka factors in different cancers, a Spearman correlation coefficient was calculated for the expression of OCT4, KLF4, SOX2, MYC, in a pan-cancer analysis that included all cancer types (Figure 5G) as well as a targeted breast cancers analysis (Figure 5H). MYC and KLF4 expression showed the most highly correlated expression profile (r = 0.35, *p* < 0.05) in the pan-cancer analysis while OCT4 and MYC expression was most positively correlated in the breast cancer analysis (r = 0.29, *p* < 0.05). These mild associations may provide clues for further investigation into pluripotency regulation within cancers and how this may be involved in contributing to susceptibility to reoviral cytolysis.

This bioinformatic analysis of resistant and sensitive cell lines brought novel leads for mechanistic insight into view. The role of PON2 in reoviral replication and cytolysis has yet to be determined. The interactions between KLF4, STAT5B and PON2 in reoviral tropism present an exciting opportunity for further assessment and may have a novel role in viral replication that has yet to be elucidated. Expression of OCT4 and MYC present an interesting avenue of investigation, particularly in basal breast cancer subtypes and may yield interesting insights into the mechanisms regulating reoviral selectivity in breast cancers. It is interesting to speculate about the relevance of these associations, as basal breast cancer are more stem-like in their characteristics, which implies these tumour types may be more responsive to reoviral therapy. Additionally, there are numerous redundant pathways that regulate pluripotency such as sonic hedgehog, notch, WNT, bone morphogenic protein, and Janus family kinase (JFK) pathways that may also contribute to reoviral selectivity but were not investigated in this study. The fact that embryonic and induced pluripotent stem cells are exquisitely sensitive to infection while their differentiated counterparts remain refractory are strong indicators that either the Yamanaka factors or additional dependent pluripotency programs are involved.

## 4. Discussion

Oncolytic viral therapies are an emerging anti-cancer strategy. Reovirus is a well-studied OV that is the only wild type virus to gain FDA approval. However, despite the initial pre-clinical promise of reovirus, results from the clinic have shown benefits for only a fraction of patients, and consequently there have been efforts to improve therapeutic efficacy [23,82,83,84,85]. The mechanisms governing reoviral tropism in cancers are not well defined and are a topic of debate. Reovirus is capable of infecting and lysing cancer cells and cancer stem cells, and, as shown here, healthy ESCs. Intriguingly, adult (mesenchymal and hematopoietic) stem cells appear to be refractory to infection [76,86]. In light of the recent view that pluripotency induction shares similar pathways with carcinogenesis, it is possible to suggest that the same subset of genes that are activated in some cancers may also be up-regulated in pluripotent cells, thereby rendering them permissive to infection. This led us to speculate that the pathways responsible for stemness in cancer and in embryonic cells also modulate reoviral susceptibility. Here we characterised the ability of reovirus to infect and kill both murine and human embryonic stem cells. We further demonstrated that reovirus can lyse induced pluripotent stem cells while their original, non-reprogrammed counterparts remained refractory. Differentiation of both embryonic and induced pluripotent stem cells altered the cells’ reoviral sensitive state to a resistant one. Additionally, murine mammary cancer cells that were non-responsive to reoviral lysis were reprogrammed using the Yamanaka factors. The newly established ipCSCs were strikingly susceptible to infection and oncolysis. Finally, we briefly investigated the potential role of the Yamanaka factors in regulating this switch through bioinformatic analysis. These data strongly suggested that stemness and the differentiation status of a cell may contribute to regulating the successful oncolytic infection of reovirus.

Viruses such as reovirus with inherent tropism for transformed cells are potentially a powerful tool that can be used to understand mechanisms of oncolytic selection, which may contribute to the development of novel cancer specific therapies. The outcome of successful viral replication and lysis is thought to be largely dependent on cellular signalling. Innumerable changes within the cell are required for oncogenesis, making it difficult to pinpoint the essential targets for successful therapies. The improvement of viruses as oncolytic agents hinges on understanding the unique set of conditions that provide a beneficial niche for viral replication. Elucidating the pathways governing reoviral selectivity will aid the development of biomarker panels for patient screening and enable the rational design of drug combination strategies for enhancing reoviral efficacy. Additionally, an understanding of the basis for tumour-selective replication could result in the discovery of new targets for small molecules and aid in the development of novel engineering strategies for OVs.

Precision medicine strategies attempt to improve the application of anti-cancer therapies by using biomarkers to predict a patient’s response to a therapy. This approach takes into account the variability of the tumour’s genetic landscape, patient characteristics and cellular environment to determine the most effective therapeutic approach to treating a specific cancer [87,88]. Establishing a panel of predictive markers for response to oncolytic viruses could help maximise clinical response and minimize off-target adverse events [88]. Oncolytics Biotech Inc. has already begun screening patients in their clinical trials based on KRAS mutation, BRAF mutation, and EGFR mutational status and amplification (NCT00861627, NCT01274624) [24]. Even so, despite intense research into reoviral replication there is no single biomarker that unambiguously indicates whether cytolytic infections will occur. This is perhaps not surprising as a multitude of signalling events must coincide to enable for productive infections. Delivery, attachment, entry, replication and apoptosis induction are all collectively required for successful oncolytic replication. Based upon the research described here, stemness may in the future be added to the list of requirements for successful infection and lysis. Indeed, Dick’s group developed a 17 gene stemness score to predict patient responses in acute leukaemia. It is therefore feasible that this type of biomarker panel can also be used to predict responses to reoviral treatment in the future [89]. Factors such as JAM-A, L and B cathepsin and β-integrin expression, RAS activation, and Yamanaka factor expression may all contribute to permissive phenotypes but not be permissive for replication in isolation [32,36,90,91,92]. A better understanding of the molecular mechanisms governing selective reoviral replication (for example, might Yamanaka factor expression modulate other pathways, such as JAM-A, cathepsins, etc.?) will aid in the development of a panel of biomarkers that can accurately and effectively be used to predict oncolytic success in patients, thereby optimising treatments.

The complexity and heterogeneity of tumours as well as their ability to develop resistance to single treatments (monotherapy) has fueled investigations into combination therapy for cancers. While oncolytic vectors are multimodal in their approach to eliminating cancers, their combination with standard treatment options and other viruses are topics of considerable research in a bid to improve their efficacy in the clinic [24,25]. Reovirus is an ideal candidate for combination therapeutics as it has minimal side effects in patients. In an attempt to rapidly translate reovirus from bench to bedside it has already been tested in combination with numerous chemotherapeutics as well as radiation therapy, thereby enhancing efficacy of both [8,18,23]. Exploration and discovery of the pathways and mechanisms responsible for reoviral selective infection will permit a more rational approach to drug combinations, allowing for the development of perhaps unexpected yet synergistic anti-cancer strategies. This may be particularly evident when considering stemness as a factor modulating reoviral tropism.

The next step in furthering the investigation into the role of pluripotency and stemness in regulating reoviral selectivity would be to infect teratomas in murine models and employ histochemical analysis and single cell sequencing to investigate the expression profile of cells within a tumour that is readily infected by reovirus. This would be fascinating in the context of a teratoma model as these tumours can produce differentiated tissue types from all three germ layers (mesoderm, ectoderm and endoderm). We earlier showed that wild type reovirus is capable of reducing the size of teratomas in NOD/SCID mice, although we did not investigate which cells within these tumours were readily infected [59]. It would be interesting to determine if differentiated tissues within the teratomas are susceptible or whether only stem-like cells within a tumour support infection and lysis. A similar approach could be taken with investigations into patient biopsies. Further investigations into the expression profiles of human cancer stem cells derived from breast tumours that have been shown to be responsive to reoviral infection may also help with this analysis [57]. The extent to which the Yamanaka factors are expressed in these cells may yield further insights into their role in regulating reoviral selection.

Although our bioinformatic analysis did not definitively elucidate the mechanism responsible for reoviral selectivity, it has provide novel avenues for investigation. In cell line analysis KLF4/STAT5B and PON2 signalling may contribute to regulating reoviral permissiveness. As KLF4 acts in concert with SOX2 and OCT4 it would be interesting to determine the combined roles of these factors in establishing reoviral susceptibility. The bioinformatic analysis of TCGA suggests that combinations of MYC and KLF4 as well as OCT4 and MYC were important for permissive cancers. It would also be interesting to determine the role of c-MYC in reoviral replication and whether it is an essential factor in regulating sensitivity. Reprogramming cells using only OCT4, SOX2 and KLF4 would be an interesting approach to testing this, as these factors can reprogram cells with reduced efficacy. Additionally, it would be intriguing to determine what role if any Nanog plays in establishing a suitable niche for reoviral replication, as stemness appears to be a key factor influencing replication. It may be of value to reprogram cells using entirely separate factors such as Lin28, Oestrogen related receptor β, Nanog, nuclear receptor subfamily 5 group A member 2 and TCL1A, all of which can promote reprograming by replacing OSKM [93]. This may be another approach to further investigate the extent to which pluripotency and stemness regulate reoviral selectivity.

Finally, if our proposal that cellular stemness modulates reoviral oncolysis is correct, we are left with the challenge of reconciling this explanation with that of Lee’s group, which earlier posited a critical role for Ras pathway activation in promoting reoviral oncolysis [6,7,8,9,10,11,12,13,14,15,16,17,18,19,20,21,22,23,24,25,26,27,28,29,30,31,32,33,34,35]. As we and others have observed (see our accompanying manuscript [94]), the correlation between Ras activation and reoviral activity is good but imperfect, with many exceptions. Thus we are led to speculate that cells able to adapt to chronic Ras-mediated signalling may become cancerous, in part, by sometimes activating stemness pathways that also confer reoviral susceptibility. We are therefore intrigued by the recent observation that Ras expression can modulate miRNA signalling, opening new possibilities for altering multiple downstream pathways, including cellular stemness [95]. In addition, an investigation of primitive gut stem cells or induced stem cells in the digestive tract that are possible natural hosts for viral infection in humans and other animals could yield further insights into mechanisms of viral cytolysis [96]. Clearly much further work would be needed to explore this or other mechanisms that may link variable Ras activation, cellular stemness, oncogenic progression and reoviral susceptibility.

## 5. Conclusions

Many factors may contribute to reoviral selectivity in cancer cells. For the first time, this study demonstrates that reprogramming differentiated cells into more stem-like progeny can make them permissive to reoviral cytolysis. Induction of stemness phenotypes promotes reoviral infection in previously resistant cell types in both healthy and cancerous cell lines. The regulatory networks involved in reprogramming have not yet been fully investigated for their involvement in regulating reoviral tropism. It is evident that the stem cell properties of cancers contribute to the most dangerous and clinically relevant aspects of cancer biology. Understanding the similarities and differences in the responses to reoviral infection could lead to the discovery of novel strategies for cancer therapeutic intervention. Further investigation into the regulatory networks within the cell that facilitate reoviral tropism are essential for advancing reovirus as an oncolytic agent. This fundamental knowledge will enable biomarker selection, rational drug combinations and ultimately contribute to enhancing the safety, specificity and potency of reoviral therapy.

## Figures and Tables

**Figure 1 viruses-15-01473-f001:**
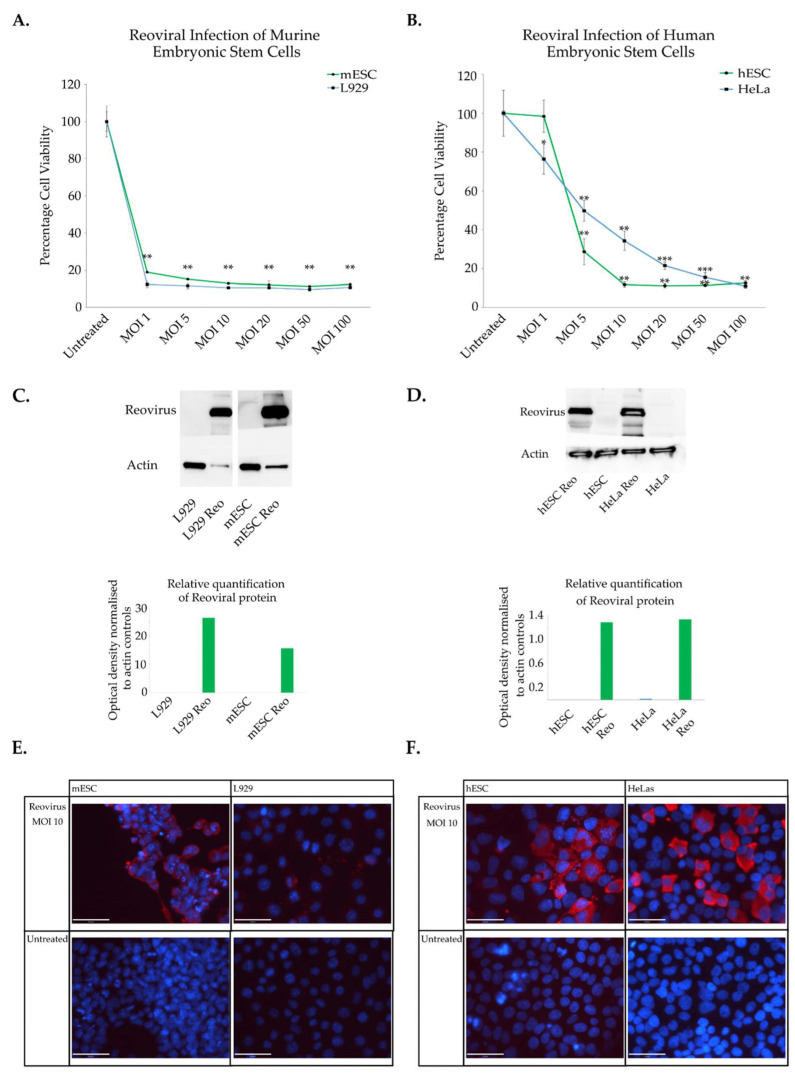
Characterisation of reoviral cytolysis and replication in cancer and embryonic stem cells. (**A**) mESCs (D3) and (**B**) hESCs (H9) cells were infected with wild type reovirus (T3D) twenty-four hours after seeding. Three days after infection cell viability was determined using crystal violet colorimetric analysis (residual staining represents remaining cell debris; all cells were killed). Absorbance was read at 570 nm and normalised to untreated controls. The assays were performed in triplicate; error bars represent standard deviation. Two-tailed paired student’s *t*-tests were used to compare treated groups to untreated controls, * *p* < 0.05; ** *p* < 0.01 and *** *p* < 0.001. Viral protein synthesis in mESCs (**C**) and hESC (**D**) was initially confirmed via Western blot. Twenty-four hours after infection with MOI 10 cells were lysed and Western blots performed with antibodies directed toward viral capsids; experiments were performed in triplicates. Relative protein quantification was measured by normalizing capsid protein expression to actin controls. Immunofluorescence imaging was also performed to assess reoviral replication. mESCs (**E**) and hESCs (**F**) were infected at an MOI 10. One day post infection cells were fixed, permeabilised and stained with antibodies directed toward viral capsids. Red—reovirus, Blue—DAPI. Scale bar is 50 μm. L929 and HeLa cells were used as positive controls. Experiments were performed in triplicates with similar results, and representative images are shown.

**Figure 2 viruses-15-01473-f002:**
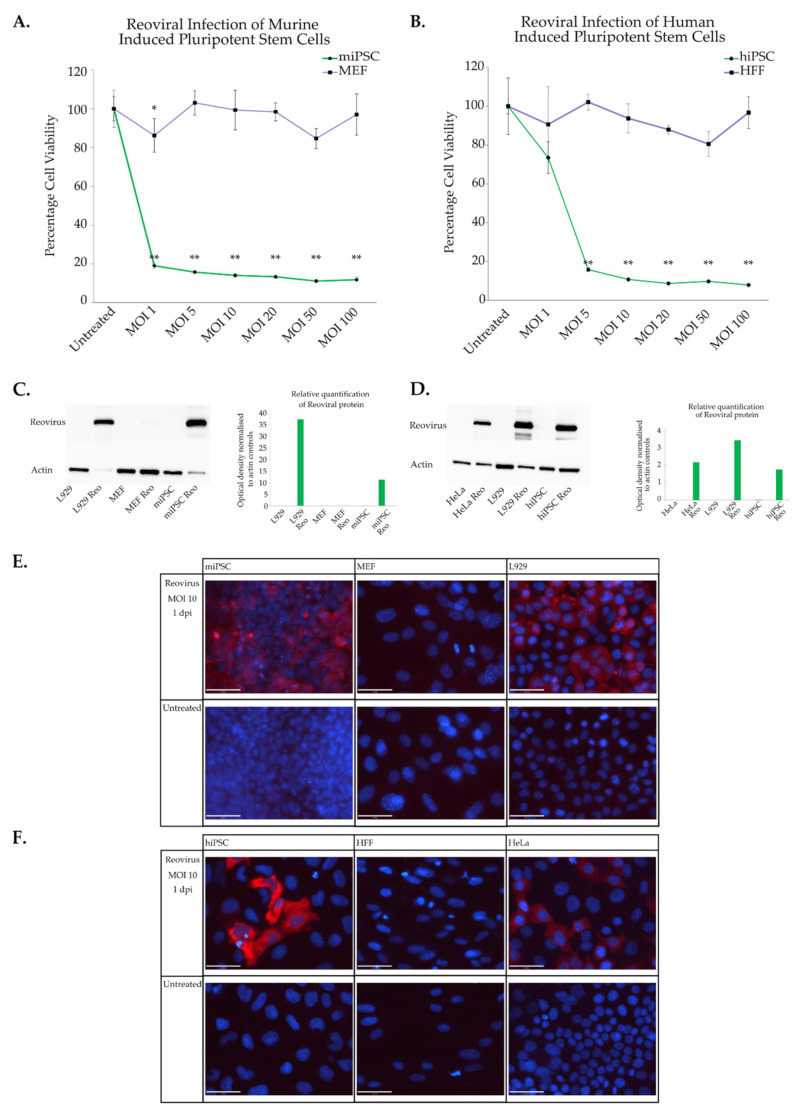
Induced pluripotent stem cell sensitivity to reoviral cytolysis and replication. (**A**) Murine differentiated cells (MEFs) and miPSCs were infected with reovirus and three days post infection were subject to crystal violet colorimetric analysis. (**B**) Similarly human foreskin fibroblasts and hiPSCs (4YA) cells were infected with wild type reovirus (T3D) twenty-four hours after seeding. Three days after infection cell viability was determined. Absorbance was read at 570 nm and normalised to untreated control. Residual staining reflects cellular debris. The assay was performed in triplicate (n = 3). Error bars represent standard deviation. Two-tailed paired student’s *t*-tests were used to compare treated groups to untreated controls, * *p* < 0.05; ** *p* < 0.01. Viral replication in miPSCs (**C**) and hiPSC (**D**) was confirmed via Western blot one day after infection (MOI 10). Relative protein quantification was measured by normalizing capsid protein expression to actin controls. Immunofluorescence imaging was performed to assess reoviral replication. One day post infection (MOI 10) miPSCs (**E**) and hiPSCs (**F**) were fixed, permeabilised and stained. Red—reovirus, Blue—DAPI. Scale bar is 50 μm. L929 and HeLa cells were used as positive controls.

**Figure 3 viruses-15-01473-f003:**
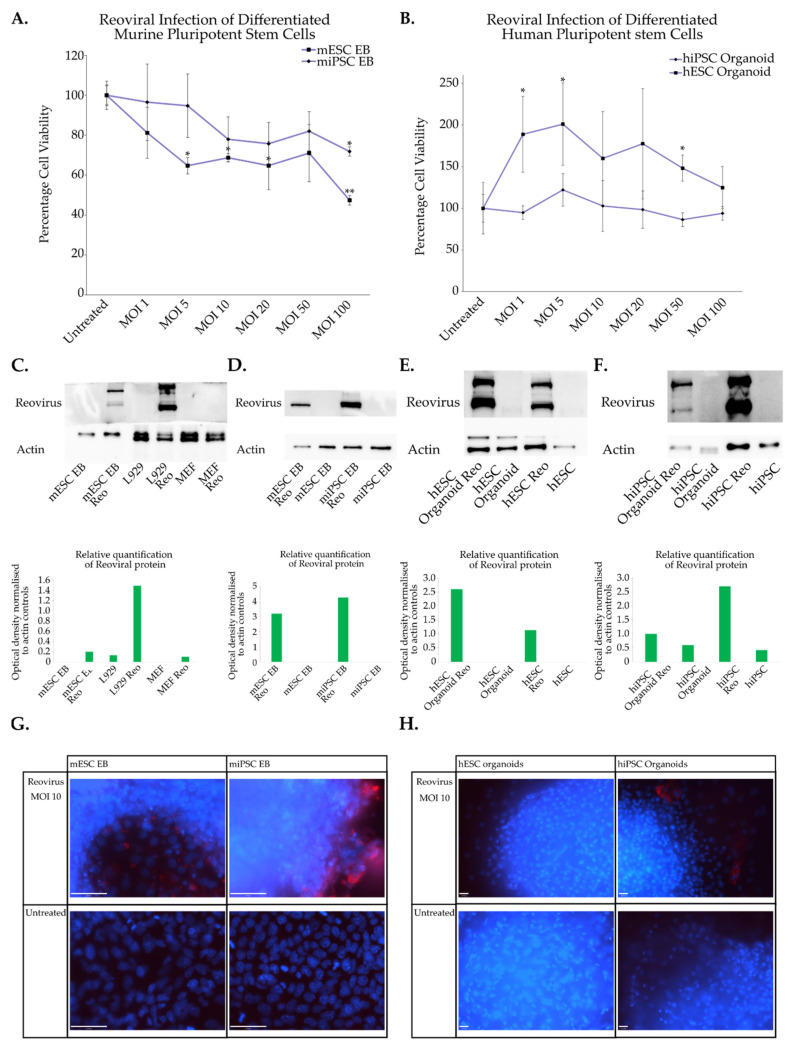
Reoviral sensitivity in differentiated stem cells. (**A**) Murine embryonic and induced pluripotent stem cells were induced to differentiate in hanging drops for three days to form embryoid bodies (EB). The cells were infected with reovirus and three days post infection crystal violet analysis was used to determine cell viability. Absorbance was read at 570 nm and normalised to untreated control. The assays were performed in triplicate (n = 3). Error bars represent standard deviation. Two-tailed paired student’s *t*-test was used to compare treated groups to untreated controls, * *p* < 0.05; ** *p* < 0.01. (**B**) Human embryonic and induced pluripotent stem cells were differentiated into liver organoids. Three days post infection cell viability was determined using crystal violet viability assay. Replicates of six (n = 6) were used for hESCs organoids (H1 organoids) and replicates of nine (n = 9) for hiPSCs organoids (4YA), * *p* < 0.05. (**C**). Error bars represent standard deviation. Two-tailed paired student’s *t*-test was used to compare treated groups to untreated controls. No biological replicates were performed for these experiments. (**C**,**D**) Murine EBs were then subjected to reoviral infection (MOI 10) for a period of 3 days and proteins were harvested and Western blots performed. (**E**,**F**) Human liver organoids were infected with reovirus (MOI 10) and three days post infection proteins was harvested for Western blot analysis. Stem cell controls were infected with MOI 10 and left for twenty-four hours before reoviral proteins were harvested. Relative protein quantification was measured by normalizing capsid protein expression to actin controls for all Western blots (**G**,**H**) Immunofluorescence imaging was performed on murine EBs and human liver organoids that had been subjected to reoviral infection (MOI 10) for two days and three days in murine and human samples respectively. Red—reovirus, Blue—DAPI. Scale bar is 50 μm.

**Figure 4 viruses-15-01473-f004:**
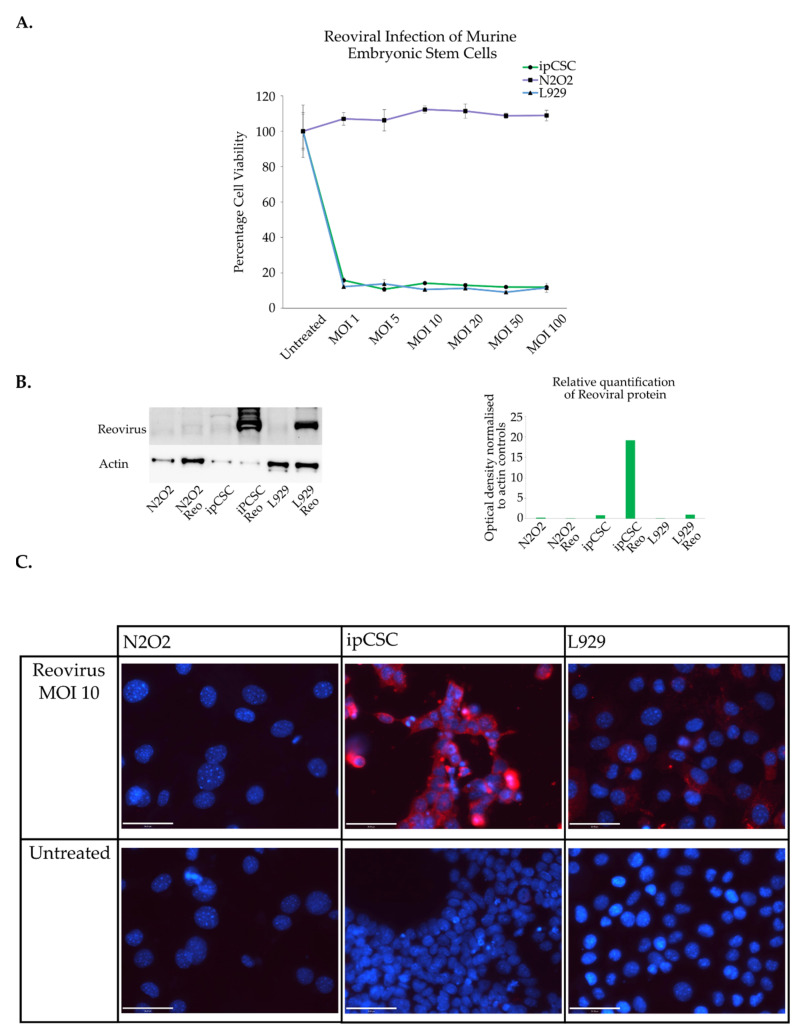
Induced pluripotent cancer stem cell sensitivity to reovirus. (**A**) Cancer cell line N_2_O_2_ derived from mammary tumours in mice and cognate ipCSC (14ex-11) were infected with reovirus at various MOIs. L929 cells were used as a positive control. Cell viability was determined using crystal violet assays three days after infection. The assay was performed in triplicate (n = 3). Absorbance was read at 570 nm and normalised to untreated control. Error bars represent standard deviation. (**B**) Western blot was conducted on proteins extracted from N_2_O_2_ and ipCSC cells three days post infection (MOI 10). Relative protein quantification was measured from the Western blots by normalising capsid protein expression to actin controls. (**C**) Immunofluorescence imaging was also performed on these cell lines one day post infection (MOI 10). Red—reovirus, Blue—DAPI. Scale bar is 50 μm.

**Figure 5 viruses-15-01473-f005:**
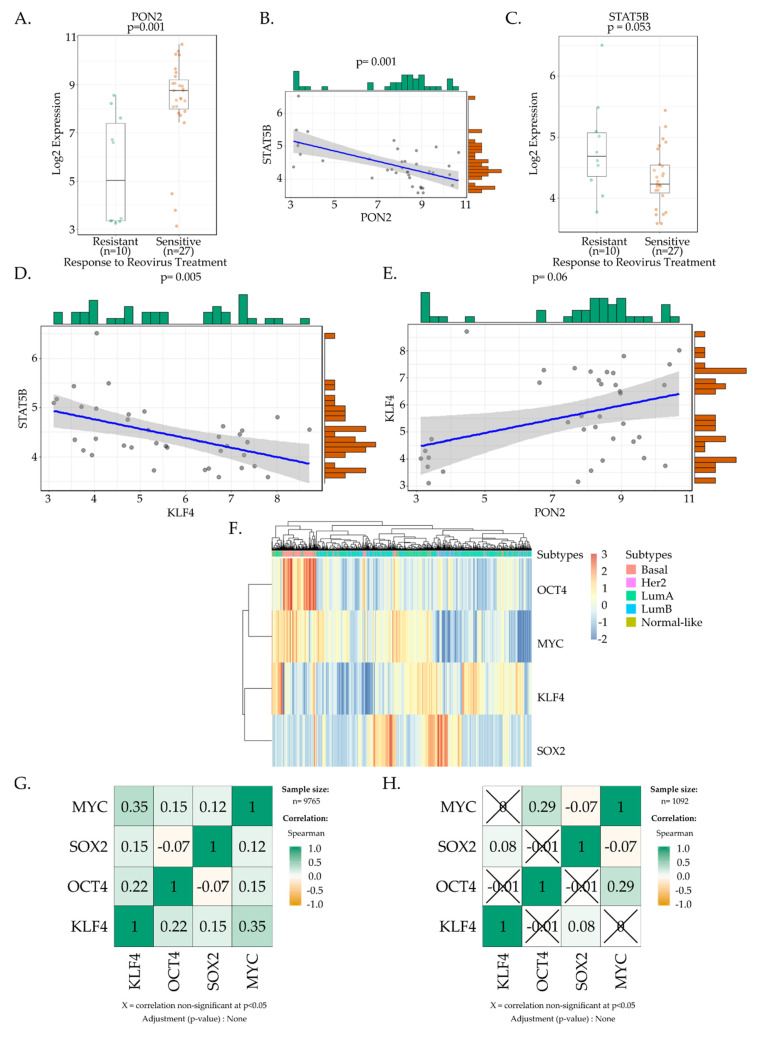
Bioinformatic analysis of Yamanaka factor expression and reoviral selectivity. (**A**) A Wilcoxon rank sum test was used to compare *PON2* expression in resistant and sensitive cell types while box plots indicate the distribution of *PON2* expression data within this dataset, *p* = 0.001. (**B**) Spearman correlation analysis was used to investigate the correlation between *PON2* and *STAT5B* expression. *STAT5B* is negatively associated with *PON2* expression in cancer cell lines *p* < 0.001. (**C**) STAT5B expression is down regulated in cancer cells that are sensitive to reoviral cytolysis. A Wilcoxon rank sum test was used to compare *STAT5B* expression in resistant and sensitive cancer cell types, *p* = 0.053. Box plots indicate the distribution of *STAT5B* expression. (**D**) To investigate the relationship between *KLF4* and *STAT5B* expression Spearman correlation analysis was performed. *KLF4* expression negatively correlates with *STAT5B* expression r = −0.45 and *p* = 0.005. (**E**) *KLF4* expression is positively associated with *PON2* expression in cancer cell lines. Spearman correlation analysis was performed and r = 0.31, *p* = 0.06. (**F**) The Heat map demonstrates the relationship between the expression levels of each of the Yamanaka factors in differing breast cancer subtypes. Expression values were log2 transformed and Z-scores normalised. Hierarchical clustering was performed using Euclidean distancing. Additional annotation tracks were included for the breast cancer subtypes. (**G**) Associations between the Yamanaka factor expression levels were analysed in a group of 30 different cancer types from the TCGA. Expression data from nearly 10 000 samples were evaluated and Spearman correlation coefficients were calculated for *OCT4, KLF4, SOX2, MYC*. *p* < 0.05. Mild associations between *KLF4* and *OCT4* as well as *KLF4* and *MYC* expression were identified. (**H**) Expression data for breast cancers were taken from the TCGA database and assessed to determine the presence of associations between Yamanaka factor expression levels. Gene expression data from 1092 samples was considered while performing Spearman correlation calculations *for OCT4*, *KLF4*, *SOX2* and *MYC*. *p* < 0.05. Mild associations between *MYC* and *OCT4* expression were noted in this cancer subtype.

## Data Availability

Data presented here may be obtained by contacting TB or RNJ.

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
