# Peer review of "Modulation of Reoviral Cytolysis (II): Cellular Stemness"

_viruses, 2023, doi:10.3390/v15071473_

Round 1

Reviewer 1 Report

This is an manuscript in which the authors built an innovative case for the involvement of the Yamanaka pluripotency factors in governing reovirus sensitivity of human and murine cells. The hypothesis was founded on the observation that human and murine undifferentiated ES and iPS cells are strongly sensitive to reovirus infection. This hypothesis may be helpful in understanding the variable therapeutic responses seen in human patients upon therapeutic reovirus administration.

The manuscript is well written and the line experimentation is well described.

Nevertheless I feel that there may be a number of issues that should be resolved.

1)      It is suggested that the wild type reovirus has ‘a natural tumor tropism’, and ‘does no harm to non-cancerous cells’. This seems an oversimplification, especially in light of the observation that most humans have been infected (as evidenced by the presence of humoral immunity against this virus) albeit most cases without any clinical manifestation. Could it be that the reovirus is selective for rapidly cycling cells, rather than for cancerous cells?

2)      Is there an effect of the Yamanaka factors on the expression of the reovirus receptors such as JAM-A? This could explain the increased sensitivity of N2O2 upon dedifferentiation. This could provide a more direct explanation of the increased susceptibility to reovirus infection upon Yamanaka-factor expression.

3)      On page 6: what is meant by ‘extensively titred’? Are all aliquots titred or all batches?

Author Response

We thank the reviewer for thoughtful and constructive comments. Our detailed responses follow:

  1. The reviewer speculates that reovirus may simply target rapidly cycling cells. However, we have many such cell populations in our bodies, and reovirus does not target them (or at least, not with clinically relevant symptoms). But it is an interesting point. We have often wondered just how reovirus can propagate in the gut of humans and many other animals if it does not harm 'normal' cells. We have no direct evidence on this point (though there are tantalizing hints in the literature), but perhaps reovirus does (weakly?) target gut stem cells, thereby causing mild diarrhea and the development of circulating antibodies. We have added a brief comment in the discussion to mention this possibility.
  2. This is another interesting idea, at least in part. Many cells express JAM-A, but only a few (embryonic stem and some cancer cells) allow productive infections leading to cell death. Thus JAM-A is a necessary but not sufficient predictor of reoviral responsiveness, as many other events (viral uncoating, replication, assembly, etc.) also need to occur. But it is possible that one or more Yamanaka factors might modulate JAM-A expression in such a way that the balance is tipped in favour of infection and cell death; we have added a comment to this effect.
  3. All batches are titred, and we have corrected this in the M&M.

Reviewer 2 Report

In this manuscript, the authors examined the impact of reovirus infection on healthy pluripotent stem cells. The last paragraph of the introduction perfectly summarizes the impact that this work can have and I perfectly agree. The use of cells (MEF and HFF) differentiated by introduction of the 4 Yamanaka factors appears as a strong and original aspect of the manuscript. I agree with the authors that: “The fact that cells become susceptible to infection upon reprogramming is a novel finding with intriguing implications.”

However, I believe that a certain number of changes are necessary to the manuscript, some aspects are not adequately presented (previous literature on reovirus and stem cells, number of replicates in some experiments, presentation of the western blots). I believe that the authors could present a much more convincing manuscript with a certain number of changes and greater attention to details.

•Introduction and literature cited: When one does a rapid PubMed search, there are a certain number of papers when stem cells were infected with reovirus. Is there a reason to disregard those? It is reasonable not to cite everything but I believe that some of these papers are of interest, but I may be in error. For example, a paper on reovirus infection of “human iPS cell-derived small intestinal epithelial-like cell” seems relevant.

•Introduction and literature cited: Although I did not verify everything, I found that some papers cited do not support the sentence where they are cited. For example: “mechanism co-ordinating RAS activation and PKR signaling in promoting reoviral replication remains unclear [36, 45, 46]” where ref 45, 46 are general references on PKR that do not seem to be reovirus-related?

•General comments on the western blots:

-Quantitation of the western blots is not very useful when looking at essentially all or nothing signals, I will remove that, especially since there is no error bars on the histograms. This raises the question, how many times were these blots done (independent experiments). Although the results are quite clear it is always better to have at least a replicate, is it the case? If so, it should be mentioned (“ the result is representative of xx independent biological replicates”).

-Also, an actin loading control is presented but the detailed protocol used is not clear, I deduce that it was done separately since I can clearly see that the exposure is different on the upper part (reovirus) and lower part (actin) of the gel. It is perfectly acceptable to perform the blot separately (or to reincubate the same membrane with a second antibody such as actin) but then the two blots should be clearly separated, better yet with lines surrounding them, now pasted together as it seems to be the case in some places, this is confusing.

•Comments on figure 5: This figure is really critical, although only based on data analysis rather than original data. It really allows an interesting analysis. However, although the authors refer to the database, a listing of the 106 cell lines and, more importantly, the 37 cell lines used in the analysis is necessary (maybe in supplementary data).

Minor points: 

•In Materials and methods, a certain number of cell lines are described but they are not actually used in the manuscript (MDA, MCF7…). It is also unusual to mention that a cell line was kindly provided by one of the coauthors. I believe that the whole Materials and methods section should be carefully edited and that everything that is described was actually used in the final version of the manuscript.

•Figure 1, panel D, should be inverted, control first then infected cells. The same is true for figure 4 panel D, E, F compared to panel C (it is actually more logical to present control uninfected cells first as in panel C).

•On page 11: It is mentioned that “Murine embryonic fibroblast (MEF) cells are a mature cell type that is derived from the skeletal muscle” I am not sure that I understand, if they are fibroblasts that are derived from connective tissue?

•The different antibodies could be presented in a Table (supplementary).

Author Response

We thank the reviewer for thoughtful and constructive comments. Our detailed responses follow:

  1. We have added a reference to intestinal stem cells and reoviral infection in the discussion.
  2. We have separated the references to indicate reovirus and PKR and general synthesis control.
  3. Regarding the Western blots, the reviewer has a keen eye for detail. The blots probed for reoviral and actin proteins were the same blots, stripped and reprobed. In assembly of the figure a misleading impression was generated in Fig 1c that the scans were continuous. We have introduced a white bar in the figure to emphasize separation of the images. We wish to retain the portion of the figure showing relative quantitation of proteins, as this shows the large differences in expression, even after normalization. This is more difficult than one might imagine, as samples are collected from infected cells that are entering apoptosis, resulting in variable degradation of proteins including actin. We loaded equal amounts of total protein, but even then actin signals could vary significantly, and thus we wished to provide our best estimates of true differences in reoviral protein abundances. We have altered the figure legends to emphasize that experiments were performed in triplicate with similar results and representative values or images are shown.
  4. Regarding Figure 5, the reviewer's request is well founded. However, we request a release from this point, as the member of our team (MK) who conducted the bioinformatics analysis and who is quite expert in this area has been hospitalized and has an uncertain recovery. He is sadly unavailable to fill in this particular gap and we are unable to communicate with him. We have copies of many intermediates in the data and figure generation process, but regretably not the exact listing of cell lines used in the analysis, only the criteria by which the list was generated. Recognizing the validity of the request, we have in partial compensation modified the text to better explain how the listing was obtained so that another expert in the field would be able to reduplicate our efforts.
  5. We have clarified in M&M that experiments were performed with all cell lines mentioned, with similar results, but only those performed in triplicate or greater were presented in the Results. We believe it is important to mention the other cell lines used (even if not reported in detail) so the readers may understand this is not a phenomenon restricted to particular cell lines, but may be generalizable, and may further serve as a guide to future experiments. We also altered the description of cells generated in the Rancourt lab, and generally 'cleaned up' the M&M section where possible.
  6. Regarding the figures, we inverted the image and labels for Figure 1 as requested. The reviewer also requests inversion for Figure 4 (but actually intends Fig 3). This would require inverting images and labels for both the Western blots and also the histograms. But we think the labels and images are in fact quite clear and unlikely to be misinterpreted by the viewer, so we request an exemption on this point.
  7. We have corrected the description of fibroblasts arising from connective tissue of murine skeletal muscle, as requested (good catch).
  8. The antibody listing has been removed from the main text and included as supplementary material (which was in fact our original intent).

Round 2

Reviewer 2 Report

No additional comment.